# Therapeutic Effect of Icaritin on Cerebral Ischemia-Reperfusion-Induced Senescence and Apoptosis in an Acute Ischemic Stroke Mouse Model

**DOI:** 10.3390/molecules27185783

**Published:** 2022-09-07

**Authors:** Cheng-Tien Wu, Ting-Hua Yang, Man-Chih Chen, Siao-Syun Guan, Chang-Mu Chen, Shing-Hwa Liu

**Affiliations:** 1Department of Nutrition, China Medical University, Taichung 406040, Taiwan; 2Master Program for Food and Drug Safety, China Medical University, Taichung 406040, Taiwan; 3Department of Otolaryngology, National Taiwan University Hospital, Taipei 10051, Taiwan; 4Institute of Toxicology, College of Medicine, National Taiwan University, Taipei 10051, Taiwan; 5Institute of Nuclear Energy Research, Atomic Energy Council, Taoyuan 32546, Taiwan; 6Division of Neurosurgery, Department of Surgery, College of Medicine and Hospital, National Taiwan University, Taipei 10051, Taiwan; 7Department of Medical Research, China Medical University Hospital, China Medical University, Taichung 406040, Taiwan; 8Department of Pediatrics, College of Medicine and Hospital, National Taiwan University, Taipei 10051, Taiwan

**Keywords:** ischemic stroke, icaritin, apoptosis, senescence

## Abstract

An ischemic stroke is brain damage caused by interruption of blood supply to the brain that can cause death and long-term disability. New medical strategies or therapies are urgently needed for ischemic stroke. Icaritin (ICT) is a metabolite of icariin (ICA), which are two active flavonoid components extracted from *Herba epimedii* and considered neuroprotective agents in animal models of Alzheimer’s disease and ischemic stroke. The therapeutic effect of ICT on ischemic still remains to be clarified. The aim of this study was to investigate the therapeutic effect of ICT on cerebral ischemia-reperfusion-associated senescence and apoptosis in a middle cerebral artery occlusion (MCAO) mouse model (ischemia for 50 min and reperfusion for 24 h). Administration of ICT after ischemia significantly reduced MCAO-induced neurological damage, infarct volume, and histopathological changes in the brain of acute ischemic stroke mice. ICT treatment could also reduce neuronal apoptosis and senescence and reversed the expression of apoptosis- and senescence-related signaling proteins. These findings suggest that ICT may have therapeutic potential to ameliorate acute ischemic stroke.

## 1. Introduction

World Health Organization data indicate that stroke is the second leading cause of death in the world, with the number of death increasing from 5.5 million in 2000 to 6.2 million in 2019 [1]. The Global Burden of Disease 2019 Study also revealed that the percentage of disability-adjusted life years (DALYs) for stroke at all ages increased from 4.2 in 1990 to 5.7 in 2019 [2]. Ischemic stroke occurs as a result of an arterial vascular blockage due to a thrombus or atherosclerosis, which is the development of fatty deposits within the vessel wall [3]. Ischemic stroke is known as the most common type of stroke, accounting for more than 80% of all strokes [3,4,5]. An ischemic stroke is a series of complex pathophysiological processes. A low rate of oxygen delivery and complete reliance on aerobic metabolism make brain tissue highly susceptible to ischemic injury [6]. Following an ischemic stroke, cerebral blood flow is disrupted throughout the affected area of the brain [7]. Currently, recombinant tissue plasminogen activator (rtPA) is the only thrombolysis drug approved by the FDA for ischemic stroke [8], which dissolves blood clots and restores blood flow in the brain. However, strict prescribing rules and potential bleeding risks are limitations that need to be overcome. Therefore, new medical strategies or therapies are urgently needed for ischemic stroke.

Icaritin (ICT) is a metabolite of icariin (ICA), both of which are the main bioactive flavonoids extracted from *Herba epimedii* (horny goat weed; Ying Yang Huo), which is a traditional Chinese medicine used to treat coronary heart disease, impotence, and osteoporosis [9,10]. Both ICA and ICT have been found to ameliorate memory and learning in experimental animal models of Alzheimer’s disease [11,12]. Zhu et al. (2010) found that treatment with ICA by intragastric administration after reperfusion in a middle cerebral artery occlusion (MCAO) mouse model protected against brain injury [13]. Xiong et al. (2016) showed that pretreatment with ICA by gavage can prevent the ischemia-reperfusion (I/R) injury in the brain in an MCAO rat model [14]. Sun et al. (2018) reported a neuroprotective effect of ICT administered by intraperitoneal injection before reperfusion on MCAO-induced brain injury in mice [15]. Recently, we found that pretreatment with both ICT and ICA administered by intraperitoneal injection before focal cerebral ischemia in a MCAO mouse model protected against neuronal injury [16]. The National Institute of Neurological Disorders and Stroke rt-PA Stroke Study group has suggested that patients who received rtPA within 3 h of the onset of ischemia exhibit improved neurological recovery and reduced disability relative to patients who received placebo treatment [17]. Therefore, the investigation of therapeutic effects of drugs on neuronal injury after acute MCAO in animal models is important. The therapeutic effect of ICT on ischemic stroke still remains to be clarified.

Torres-Querol et al. recently showed that a cellular senescence-associated secretory phenotype plays a role in acute ischemic stroke in a transient focal cerebral ischemia mouse model [18]. A recent study reported that the cellular senescence induced by stroke contributes to neurological function loss in a MCAO rat model [19]. These findings imply that brain cell senescence may be an important pathological mechanism of acute ischemic stroke.

The aim of this study was to investigate the therapeutic effect of ICT administered after acute MCAO-induced ischemia in mice on cerebral ischemia-reperfusion-associated neuronal injury. Neuronal functions, brain infarct volume, and histopathological changes were evaluated. Neuronal apoptosis and senescence, as well as apoptosis- and senescence-related signaling proteins in the cortex and hippocampus, were also examined.

## 2. Results and Discussion

In this study, we testes the therapeutic effect of ICT administered after ischemia on acute ischemic stroke in mice. An experimental flow chart of the therapeutic effect exerted by ICT in an acute cerebral ischemia/reperfusion injury mouse model is shown in Figure 1A. 

### 2.1. Effects of ICT Treatment on Neurological Functions and Brain Pathological Changes in Acute Ischemic Stroke Mice

Body weight loss, mNSS, and brain infarct volume were observed to explore the therapeutic effect of ICT on mice with acute brain I/R injury. As shown in Figure 1B, ICT and edaravone (positive control) did not affect the percent weight loss induced by MCAO surgery. Next, the degree of neurological damage induced by ischemia-reperfusion was assessed using mNSS. As shown in Figure 1C, the mNSS in the I/R group was expressed as moderate damage (mean score, 10.2), whereas the mNSS in the I/R+ICT and I/R+E groups was expressed as mild damage (mean score, 4.3 and 2.8, respectively). Therefore, ICT and edaravone treatment can significantly reduce the severe nerve damage caused by MCAO surgery. In addition, TTC staining was used to assess ischemia-reperfusion-induced infarct volume. TTC is hydrophilic and photosensitizing and can differentiate between fresh tissue and infarcted tissue [20]. As shown in Figure 1D,E, the infarct volume was significantly reduced in both the I/R+ICT and I/R+E groups compared to the I/R group. All these results suggest that the therapeutic effect of ICT can reduce nerve damage, as well as cerebral infarct volume, after MCAO surgery in an acute brain I/R injury mouse model.

After I/R, ischemia-injured neuronal cells typically exhibit irregular atrophy and eosinophilic cytoplasm, shrunken nucleus with pyknosis (irreversible condensation of chromatin), and TUNEL-positive staining [21]. As shown in Figure 2, the I/R group had marked neuronal atrophy and eosinophilic cytoplasm in the cortex and hippocampus tissues. ICT treatment effectively alleviated the morphological changes of the ischemic brain. 

In our previous study, ICT pretreatment before ischemia was found to significantly decrease I/R-induced body weight loss in mice [16], whereas in the current study, ICT treatment after ischemia resulted in no significant change in body weight loss. Moreover, the average mNSS score and infarct volume percentage were 3.25 and 7.23%, respectively, in ICT pretreatment I/R mice (11.5 and 62.4%, respectively, in the I/R group) [16], whereas in the present study, the average mNSS score and infarct volume percentage were 4.3 and 9.1%, respectively, in ICT treatment I/R mice (10.5 and 59.3%, respectively, in the I/R group). These results indicate that the ICT pretreatment condition may have a more effective protective effect against ischemic stroke than the treatment/therapeutic condition. However, comprehensive observation of neurological function indicators and brain infarct volume, as well as histopathological analysis, showed that ICT treatment after ischemia still has a significant effect in terms of relieving acute ischemic stroke. 

### 2.2. Effects of ICT treatment on Cerebral Apoptosis and Senescence in Acute Ischemic Stroke Mice

To assess the number of apoptotic cells in brain sections, TUNEL staining was used to detect fragmented DNA of apoptotic cells. TUNEL-positive cells were stained green, whereas nuclei were stained blue with DAPI (Figure 3). In the I/R group, TUNEL-positive cells were present in both the hippocampus and cortex tissues (Figure 3). ICT treatment effectively reduced the number of TUNEL-positive cells compared to the I/R group (Figure 3). These results suggest that ICT treatment can reduce I/R-induced neuronal injury, as well as histopathological changes and apoptosis.

Apoptosis is known to play an important role in acute ischemic stroke [22]. We further collected cortical and hippocampal tissues separately for apoptosis-related protein expression analysis. The levels of protein expression for cleaved caspase3, cleaved PARP, and Bax were markedly increased, and protein expression for Bcl-2 was significantly decreased in the cortical and hippocampal tissues of I/R mice (Figure 4). ICT processing significantly reversed the protein expression levels of these proteins in the cortical and hippocampal tissues (Figure 4). These findings suggest that ICT has a protective effect against ischemia-reperfusion-induced apoptosis in the brain.

Antiapoptotic Bcl-2 family and proapoptotic proteins play important roles in the apoptosis signaling pathway. Antiapoptotic proteins can promote cell survival through a variety of mechanisms, including by modulating the activation of caspase cascades and preventing the release of apoptotic proteins [23]. In vivo studies have also shown that Bcl-2 overexpression reduces I/R-induced infarct volume, whereas Bcl2 knockout in mice exacerbates I/R-induced brain damage [24,25]. In the present study, we found that ICT treatment after ischemia effectively reversed the I/R-induced apoptosis and apoptosis-related signaling proteins in the brain.

Cellular senescence is often associated with cell cycle arrest triggered by various kinds of damage, such as DNA damage. Major cellular senescence factors include elevated levels of p21, p27, p53, and senescence-associated β-galactosidase (SA-β-gal) proteins. p21 and p27 are cyclin-dependent kinase inhibitors responsible for the senescent cell phenotype [26]. SA-β-gal is a common and long-established cellular biomarker of cellular senescence [27]. A study involving human skin from donors of various ages revealed age-dependent increase in SA-β-gal in dermal fibroblasts and epidermal keratinocytes [27]. As shown in Figure 5, the content of SA-β-gal was higher in the hippocampus and cortex of I/R group mice, which was effectively reversed by ICT treatment. We further explored the protein expression of senescence markers p53, p27, and p21. As shown in Figure 6, p53, p27, and p21 were highly expressed in the hippocampal tissues after I/R, which could be significantly reversed by ICT treatment. In the cortical tissues, the protein expression levels of p27 and p21, but not p53, were significantly increased in the I/R group, which could be reversed by ICT. These results suggest that ICT may have the ability to reduce ischemia-reperfusion-induced neuronal senescence. 

The activation of the p53-p21 pathway is known to play an important role in regulating cell senescence [28]. This pathway has been shown to be involved in tubular cell senescence after renal ischemia-reperfusion injury [29]. Cheng et al. found that the p53 and p21 signaling pathway participates in cellular senescence in the mouse hippocampus after whole-brain irradiation [30]. The upregulation of p27 can also contribute to cellular senescence [31,32]. In the present study, we found that ICT treatment after ischemia effectively reduced cellular senescence and senescence-related signaling proteins in the brain.

ICT is known to possess the anti-inflammatory and antioxidative properties and to have neuroprotective potential with respect to various insult-induced neuronal injuries in vitro and in vivo [15,16,33,34,35]. ICT has also been shown to exert preventive effects on neuronal injury in ischemic stroke mouse models [15,16]. However, the therapeutic effect of ICT on ischemic stroke still remains to be clarified. In this study, we demonstrated that ICT treatment after ischemia effectively reduced cerebral ischemia-reperfusion-associated senescence and apoptosis in an acute ischemic stroke mouse model.

This study is subject to some limitations. Changes at the metabolite and protein level can be investigated using a metabolomic and proteomic approach in the future to support the development of the proposed ICT therapy approach by simultaneously examining potential changes in multiple biochemical pathways.

## 3. Materials and Methods

### 3.1. Middle Cerebral Artery Occlusion (MCAO) Mouse Model

Male ICR mice (4–5-week-old) provided by the Laboratory Animal Center, College of Medicine, National Taiwan University (NTU; Taipei, Taiwan), were housed in an environment with controlled temperature and 12 h light/dark cycle and provided adequate food and water ad libitum. The procedures of animal experiments followed a protocol (No. 20190111) and guidelines approved by the Animal Research Committee, College of Medicine, NTU. Mice were randomized into four groups: (1) sham control, (2) MCAO (I/R), (3) I/R + edaravone (3 mg/kg in dimethyl sulfoxide (DMSO); Selleck Chemicals, Houston, TX, USA), and (4) I/R + ICT (60 mg/kg in DMSO). Edaravone treatment was used as a positive control [36]. ICT was purchased from Cayman Chemical (Cat. No. Cay20236-500, >98% purity; Ann Arbor, MI, USA). For MCAO surgery, an intermediate incision was performed in the neck of mice after anesthesia by isoflurane (Tokyo Chemical Industries, Ltd., Tokyo, Japan) + 3% oxygen. Then, the left common carotid artery was isolated, and a 6-0 nylon thread was inserted from the external carotid artery incision until the middle cerebral artery was occluded. The ischemia period was 50 min. The cerebral blood flow in MCAO mice was monitored by a laser Doppler device (PeriFlux 4001, Perimed, Stockholm, Sweden) [16,37]. In the sham group, the same procedure was performed without inserting a nylon thread. Both edaravone and ICT were administered by intraperitoneal injection (i.p.) after MCAO ischemia. Then, 24 h after reperfusion, all mice were euthanized and necropsied. During and after MCAO surgery, the rectal temperature of mice was maintained at 37.0 °C using a temperature-controlled heating pad. Dose selection for edaravone and ICT was based on the previous studies [16,38] and preliminary trials.

### 3.2. Neurological Score Evaluation

We used the modified neurological severity scale (mNSS) to assess neurological damage due to stroke. A 0–14 scale (normal score = 0 and maximum deficit score = 14) was used to score the mNSS and detect motor, sensory, reflex, and balance behaviors. Li et al. modified the assessment of neural functions in mice [39]. The scoring scale is based on the severity of the neurological injury: severe injury, 10–14; moderate injury, 6–10; and mild injury, 1–5.

### 3.3. Determination of Infarct Volume and Histopathological and Immunohistochemical Changes

Brains were immediately removed from euthanized mice and cut into 2 mm thick coronal slices after assessment of nerve injury. Brain sections were then stained with 2% 2,3,5-triphenyltetrazolium chloride (TTC; Sigma-Aldrich, St. Louis, MO, USA) and incubated at 37 °C for 20 min. TTCs can be converted to red color by mitochondria in a viable brain tissue, whereas the infarct areas are colorless. The TTC-stained sections and infarct volumes were then measured after being photographed. ImageJ software [40] was used to analyze the infarct volumes. The areas of infarct were summed and divided by the total volume of the slice and displayed as a percentage of the contralateral hemisphere volume.

After mice were euthanized, brain samples were isolated and fixed as previously described [37]. Fixed tissues were embedded in paraffin, and 4 μm tissue sections were prepared. Hematoxylin and eosin (H&E)-stained tissue sections were used to analyze histopathological changes.

For IHC analysis, brain sample sections were deparaffinized by immersing them in xylene substitute overnight. After rehydration, hydrogen peroxidase block (ab127055, Abcam, Cambridge, MA, USA) and protease (Sigma-Aldrich, St. Louis, MO, USA) solutions were added for 10 min to remove the interference of endogenous peroxidase activity. Next, protein blocks were added for 10 min to prevent non-specific binding of antibodies. A primary antibody (anti-β-galactosidase (ab4761, 1:500, Abcam, Cambridge, MA, USA)) was then added and incubated at 4 °C overnight. The following day, the primary antibody was conjugated with biotinylated goat anti-polyvalent, streptavidin peroxidase and diaminobenzidine (DAB) chromogen for 10 min. Tissue sections were then counterstained with hematoxylin.

### 3.4. Terminal Deoxynucleotidyl Transferase (TdT) dUTP Nick End Labeling (TUNEL) Analysis

A TUNEL assay determined by a DeadEnd^TM^ fluorometric TUNEL system (Promega Corporation, Madison, WI, USA) was used to detect DNA fragments of late apoptotic cells as previously described [39]. Briefly, brain tissue sections were deparaffinized by immersion in xylene substitute and rehydrated with reduced-strength ethanol/saline buffer with 0.85% NaCl. Tissue sections, which were previously fixed with paraformaldehyde for 15 min, were then incubated in 100 µL of TdT incubation buffer for 1 h in a dark humidity chamber at 37 °C. Sections were counterstained with mounting medium containing 4′,6-diamidino-2-phenylindole (DAPI; Sigma-Aldrich, St. Louis, MO, USA). Finally, fluorescein-12-dUTP-labeled DNA was visualized using a fluorescence microscope.

### 3.5. Immunoblot Analysis

To investigate the effects of ischemia-reperfusion injury on specific regions of the brain, we collected the cortex and hippocampus separately for protein expression analysis. Western blot analysis was performed as previously described [41]. These tissues were collected and lysed with radioimmunoprecipitation (RIPA) buffer (Tris-HCl (pH 7.4), 20 mM; NaCl, 150 mM; EDTA, 1 mM; ethylene glycol tetraacetic acid, 1 mM; Nonidet P-40, 0.1%; protease and phosphatase inhibitors, 1 µg/mL (Thermo Fisher Scientific, Waltham, MA, USA)) overnight at −20 °C and centrifuged at 13,000 rpm for 30 min. To obtain equal amounts of protein, the protein concentration was determined using a Pierce bicinchoninic acid (BCA) protein detection kit (Thermo Fisher Scientific, Waltham, MA, USA) in the supernatant. Subsequently, the supernatant containing the SDS buffer was heated at 95 °C for 10 min at equal concentrations (10–20 µg) to denature the proteins. Next, protein samples were separated using 10–15% sodium dodecyl sulfate polyacrylamide gel electrophoresis (SDS-PAGE) at 80–120 V and blotted to polyvinylidene fluoride (PVDF; Millipore, Burlington, MA, USA). PVDF membranes were blocked with 5% nonfat milk dissolved in 0.1% TBST (Tris-HCl, pH 7.5, 50 mM; NaCl, 150 mM; Tween 20, 0.1%) buffer for 1 h. The primary antibodies were added and incubated with membrane at 4 °C overnight. The membrane was washed three times with 1% TBST for 10 min each time; then, the membrane was incubated with HRP-conjugated secondary antibody for 1 h at room temperature. Protein bands were enhanced by chemiluminescent reagents (Bio-Rad, Hercules, CA, USA). Protein expression levels were quantitatively assessed by densitometry. ImageJ analysis software [40] was used to quantify protein expression normalized by β-actin.

### 3.6. Statistical Analysis

Data are expressed as the mean ± S.D. of at least three independent experiments. One-way analysis of variance (ANOVA) was performed, followed by Tukey’s post hoc test to determine the statistical significance among groups. A significance threshold of *p*-value < 0.05 was determined by GraphPad Prism (San Diego, CA, USA).

## 4. Conclusions

In the present study, we found that ICT treatment after ischemia significantly decreases I/R-induced cellular apoptosis and senescence in the cortex and hippocampus in an acute ischemia stroke mouse model. These findings suggest that ICT may have therapeutic potential to ameliorate acute ischemic stroke.

## Figures and Tables

**Figure 1 molecules-27-05783-f001:**
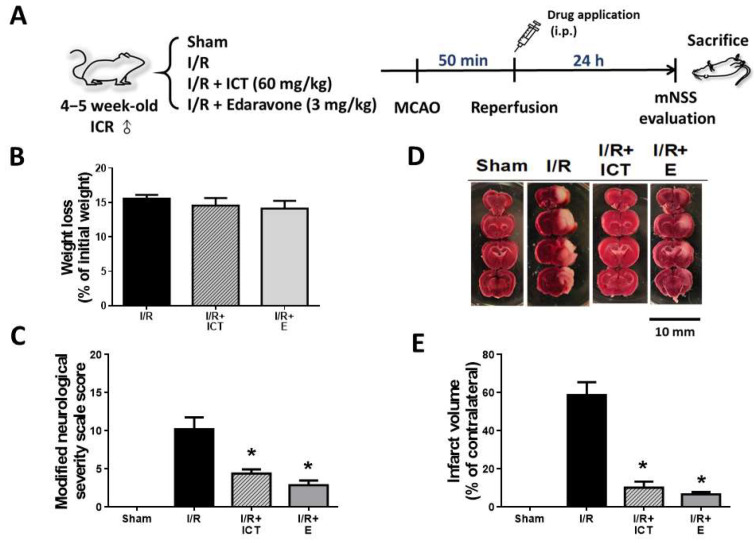
The therapeutic effects of icaritin (ICT) on body weight loss, neurological severity, and infarct volume in mice with acute cerebral ischemia-reperfusion. (**A**) Experimental flow chart of the therapeutic effect exerted by ICT in an acute cerebral ischemia/reperfusion injury mouse model. (**B**–**E**) Mice were treated with ICT (60 mg/kg) and edaravone ((**E**); 3 mg/kg, as a positive control) after ischemia. The average body weight loss percentage of mice in each group was calculated (**B**). The modified neurological severity score (mNSS) in each group was evaluated (**C**). Photographs of the mouse cerebral infarct areas in each group (**D**). Quantification of infarct volume (**E**). Data are presented as mean ± SD (n = 6). * *p* < 0.05 compared to the MCAO group.

**Figure 2 molecules-27-05783-f002:**
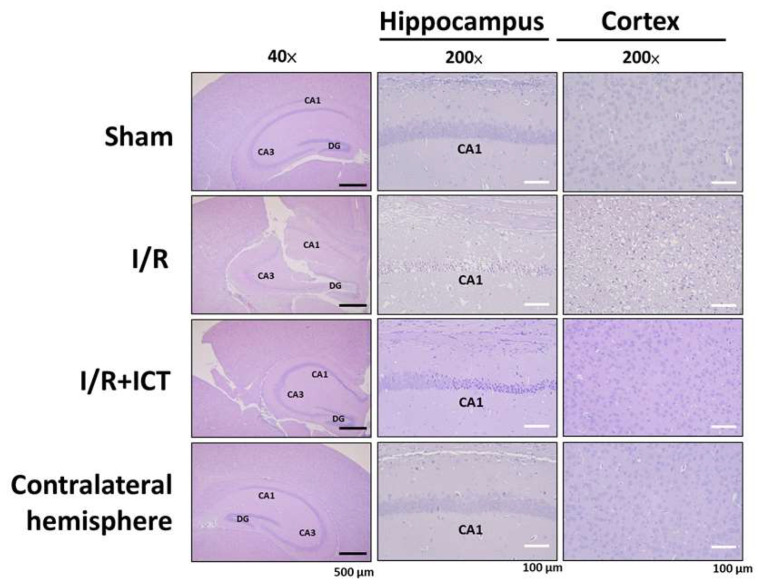
Effects of ICT treatment on histopathological changes in the left cerebral hemisphere of mice with acute cerebral ischemia-reperfusion. H&E staining was used to detect histopathological changes in both the cortex and hippocampus under 40× and 200× magnification. Black scale bar = 100 μm; white scale bar = 500 μm.

**Figure 3 molecules-27-05783-f003:**
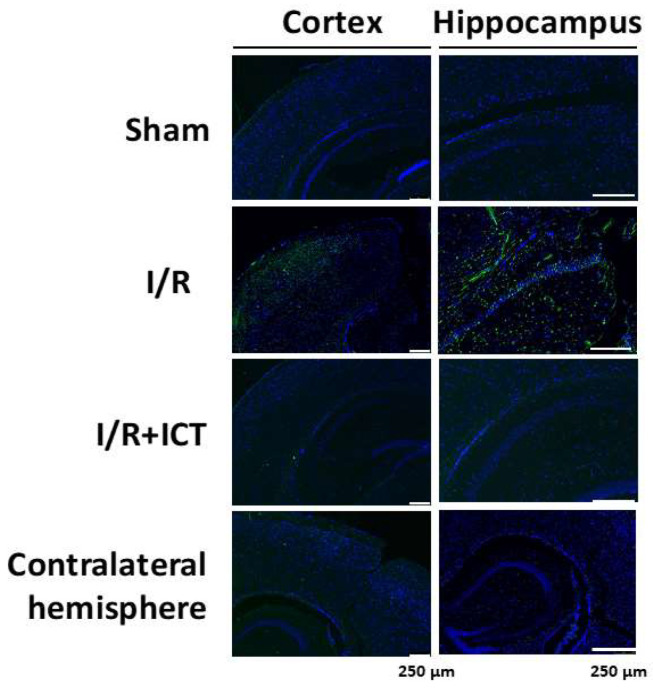
Effects of ICT treatment on neuronal cell apoptosis in the left cerebral hemisphere of mice with acute cerebral ischemia-reperfusion. TUNEL staining was used to detect neuronal apoptotic cells in the hippocampus and cortex. TUNEL-positive cells were stained a fluorescent green color, whereas cell nuclei were stained a fluorescent blue color. Scale bar = 75 μm.

**Figure 4 molecules-27-05783-f004:**
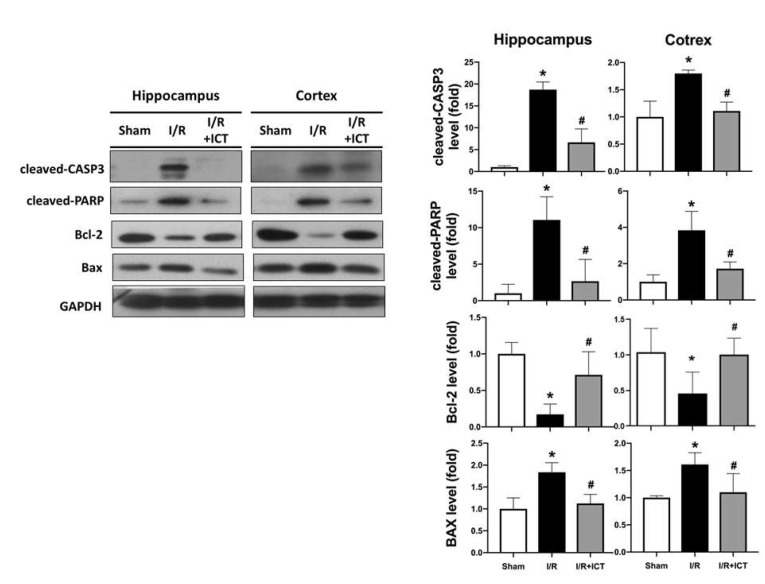
Effects of ICT treatment on the levels of protein expression of apoptotic markers in the left cerebral hemisphere of mice with acute cerebral ischemia-reperfusion. The levels of protein expression of apoptotic markers (cleaved caspase-3, cleaved PARP, Bcl-2, and Bax) in the hippocampus and cortex areas were determined by Western blotting. Protein expression was quantified by densitometry. Data are presented as mean ± SD (n ≥ 4). * *p* < 0.05 compared to sham group; # *p* < 0.05 compared to I/R group.

**Figure 5 molecules-27-05783-f005:**
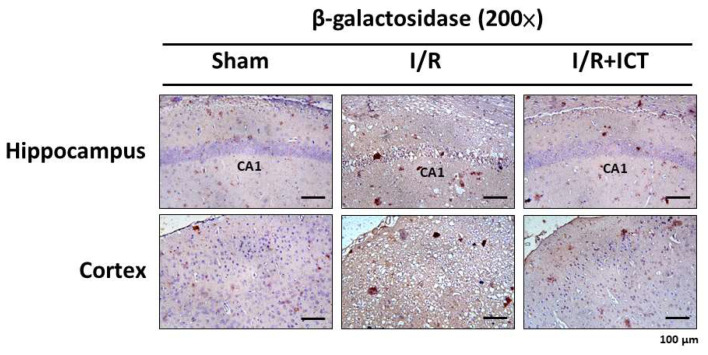
Effects of ICT treatment on the expression of senescence markers induced by MCAO in the cerebral hippocampus and cortex of mice. Immunohistochemical images of the expression of senescence-associated β-galactosidase (SA-β-gal) in the hippocampus and cortex of mouse brain sections. Images scale bar = 100 µm.

**Figure 6 molecules-27-05783-f006:**
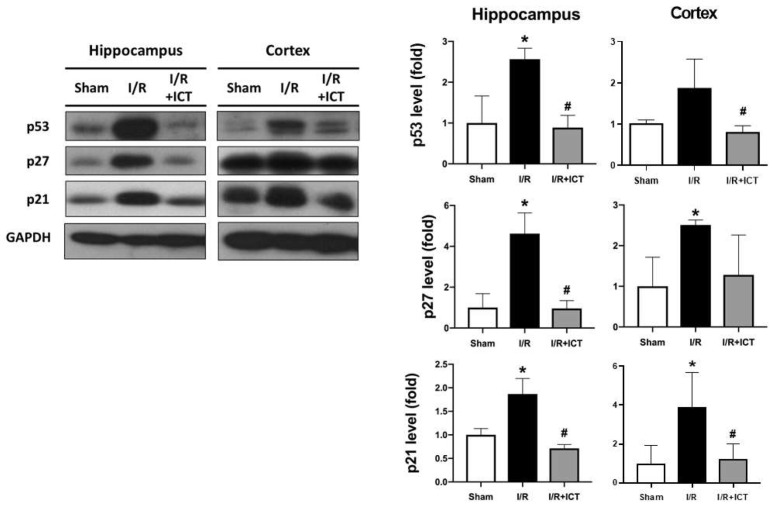
Effects of ICT treatment on cell senescence in mouse brain tissue induced by ischemia-reperfusion. Senescence-associated proteins (p53, p27, and p21) were determined by Western blotting. Data are presented as mean ± SD (n ≥ 4). * *p* < 0.05 compared with sham group; # *p* < 0.05 as with I/R group.

## Data Availability

The data presented in this study are available from the corresponding author upon reasonable request.

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
