# Peer review of "Therapeutic Effect of Icaritin on Cerebral Ischemia-Reperfusion-Induced Senescence and Apoptosis in an Acute Ischemic Stroke Mouse Model"

_molecules, 2022, doi:10.3390/molecules27185783_

Round 1

Reviewer 1 Report

The topic presented in the article is very interesting. New compounds and new therapies are essential for treating post-ischemic stroke patients. The article is very well prepared and I found only a few minor flaws in the materials and methods.

In my opinion, the authors should include the results obtained with the LC-MS/NMR technique in subsequent papers. It can be very interesting to observe changes at the level of metabolites and proteins using a metabolomic and proteomic approach. This is likely to help develop this therapy faster by examining potential changes in multiple biochemical pathways simultaneously.

Besides, authors should check and correct the materials and methods part - visible minor errors in the text.  

Author Response

Reviewer #1

The topic presented in the article is very interesting. New compounds and new therapies are essential for treating post-ischemic stroke patients. The article is very well prepared and I found only a few minor flaws in the materials and methods.
Response: We appreciate the reviewer's positive comment.

In my opinion, the authors should include the results obtained with the LC-MS/NMR technique in subsequent papers. It can be very interesting to observe changes at the level of metabolites and proteins using a metabolomic and proteomic approach. This is likely to help develop this therapy faster by examining potential changes in multiple biochemical pathways simultaneously.

Response: We appreciate the reviewer's comment. We added a statement as a limitation of our study and the possible future work item for this issue. We added the descriptions about the metabolomic and proteomic approach in the Discussion section of this revised manuscript according the suggestion of reviewer.  The descriptions were shown in the text as follows: "This study still has its limitation. It can be investigated to observe changes at the levels of metabolites and proteins using a metabolomic and proteomic approach in the future that may be likely to help develop this ICT therapy approach faster by examining potential changes in multiple biochemical pathways simultaneously."

Besides, authors should check and correct the materials and methods part - visible minor errors in the text.  

Response: We appreciate the reviewer's comment. We have checked and revised the visible errors in the text of this revised manuscript according to the suggestion of reviewer.

Reviewer 2 Report

This manuscript by Wu and colleagues continues their previous study that Icaritin protects against ischemia stroke-induced apoptosis, and offers new findings on Icaritin against ischemia stroke-induced senescence in brain. This is a well-organized study, with an easy-to-follow flow of ideas. This reviewer has only one comment before it can be published.

1.      Using Icaritin to treat ischemia stroke-induced senescence appears very abruptly in the results. The Reviewer suggest adding some description to Introduction about ischemia stroke triggering cellular senescence or the reasons of investigating Icaritin treatment cellular senescence.

Author Response

Reviewer #2

This manuscript by Wu and colleagues continues their previous study that Icaritin protects against ischemia stroke-induced apoptosis, and offers new findings on Icaritin against ischemia stroke-induced senescence in brain. This is a well-organized study, with an easy-to-follow flow of ideas. 

Response: We appreciate the reviewer's positive comment.

This reviewer has only one comment before it can be published.

  1. Using Icaritin to treat ischemia stroke-induced senescence appears very abruptly in the results. The Reviewer suggest adding some description to Introduction about ischemia stroke triggering cellular senescence or the reasons of investigating Icaritin treatment cellular senescence.

Response: We appreciate the reviewer's comment. We have added the descriptions about ischemia stroke triggering cellular senescence or the reasons of investigating Icaritin treatment cellular senescence in the Introduction of this revised manuscript according to the suggestion of reviewer. The descriptions were shown in the text as follows: "Torres-Querol et al. have recently shown that a cellular senescence-associated secretory phenotype plays a role in acute ischemic stroke in a transient focal cerebral ischemia mouse model [18]. A recent study has also reported that the cellular senescence induced by stroke contributes to the neurological function loss in a MCAO rat model [19]. These findings imply that the brain cell senescence may be an importantly patho-logical mechanism for acute ischemic stroke."